# Antenna Model with Pattern Optimization Based on Genetic Algorithm for Satellite-Based SAR Mission

**DOI:** 10.3390/s25154835

**Published:** 2025-08-06

**Authors:** Saray Sánchez-Sevilleja, Marcos García-Rodríguez, José Luis Masa-Campos, Juan Manuel Cuerda-Muñoz

**Affiliations:** 1INTA National Institute of Aerospace Technology, Ajalvir Road, km 4, 28850 Torrejón de Ardoz, Spain; ssansev@inta.es (S.S.-S.); garciarm@inta.es (M.G.-R.); cuerdamjm@inta.es (J.M.C.-M.); 2E.T.S.I. Telecommunication, Department of Signals, Systems and Radiocommunications, Universidad Politécnica de Madrid (UPM), C/Ramiro de Maeztu 7, 28040 Madrid, Spain

**Keywords:** satellite-based SAR, phased arrays, beams, excitations, optimization, genetic algorithm, antenna model

## Abstract

Synthetic aperture radar (SAR) systems are of paramount importance to remote sensing applications, including Earth observation and environmental monitoring. Accurate calibration of these systems is imperative to ensuring the accuracy and reliability of the acquired data. A central component of the calibration process is the antenna model, which serves as a fundamental reference for characterizing the radiation pattern, gain, and overall performance of SAR systems. The present paper sets out to describe the implementation and validation of a phased-array antenna model for Synthetic Aperture Radar Systems (SARAS) in MATLAB R2024a. The antenna model was developed for utilization in the Spanish Earth observation missions PAZ and PRECURSOR-ECO. The antenna model incorporates a number of functions, which are divided into two primary modules: the first of these is the antenna pattern generation (APG) module, and the second is the antenna excitation generation (AEG) module. The present document focuses on the AEG, the function of which is to generate patterns for all required beams. These patterns are optimized and matched to specific calculated masks using an ad hoc genetic algorithm (GA). In consideration of the aforementioned factors, the AEG module generates a set of complex excitations corresponding to the required beam from different satellite operational beams, based on several radiometrically defined parameters.

## 1. Introduction

Synthetic aperture radar (SAR) is a form of active remote sensing that utilizes the motion of a radar antenna over a target region to simulate a large virtual aperture. This enables the generation of high-resolution two- and three-dimensional images, regardless of the weather conditions or lighting [1,2]. In contradistinction to optical sensors, SAR systems utilize the emission of microwave signals, subsequently recording the backscattered echoes from the Earth’s surface. This renders them highly effective for a range of applications, including topographic mapping, land use monitoring, disaster assessment, and military surveillance [3,4]. The capacity of SAR to function in all-weather, day-and-night conditions, along with its ability to perform interferometric and polarimetric measurements, has rendered it an indispensable tool in both civilian and defense domains [4]. Recent advancements in spaceborne and airborne platforms have enhanced the temporal and spatial resolution of SAR, thereby expanding its utility across a range of geospatial disciplines.

The utilization of actual SAR satellites facilitates a multitude of acquisition modes, including but not limited to Stripmap, ScanSAR, and other wide-swath and high-resolution modes. Consequently, a plethora of antenna beams and, by extension, antenna patterns are required to ensure precise calibration of the modes against each other and within the SAR image itself. The generation of images with a high resolution in the meter range is a key capability, as is the accurate measurement of backscatter down to a few tenths of dB [5].

As these points demonstrate, accurate knowledge of the antenna patterns in synthetic aperture radar (SAR) is of principal importance for the purpose of processing SAR images with a high degree of precision. In this particular context, the most significant element is the antenna model approach that has been outlined in this paper. The model in question derives the antenna patterns from a combination of mathematical models and on-ground measurements. These on-ground measurements characterize parts of the array antenna. This approach facilitates the calibration of a substantial number of used antenna beams with both high accuracy and efficiency, as well as a reduced time and cost [5,6]. Furthermore, it is imperative to acknowledge the significant impact of SAR antenna patterns on system performance. The main lobe beam pattern generated by the SAR antenna is related to the noise-equivalent sigma zero (NESZ) and the swath width. In contrast, the sidelobe beam pattern is related to the azimuth-ambiguity-to-signal ratio (AASR) and the range-ambiguity-to-signal ratio (RASR). As the pulse repetition frequency (PRF) increases, the azimuth ambiguity in the along-track direction decreases, whereas the range ambiguity in the cross-track direction increases [1,6,7]. In consideration of the trade-off between the PRF, AASR, and RASR, the focus of previous studies has been on the design method. The objective of this method is to improve the RASR performance by optimizing the elevation beam patterns under relatively high-PRF conditions to secure the AASR performance [8,9,10,11,12,13,14].

Taking this into account, this paper describes the development, implementation, and validation of a novel modular antenna modeling framework in MATLAB R2024a, named SARAS (SAR Array Synthesis). SARAS consists of two principal modules: antenna pattern generation (APG) and antenna excitation generation (AEG). Both modules include user-friendly graphical interfaces that allow for comprehensive configuration of the antenna and mission parameters. Thanks to its modular and customizable architecture, SARAS is applicable not only to the PAZ and PRECURSOR-ECO Spanish missions but also to a wide variety of SAR satellite systems.

A key innovation introduced in this work is the integration of a custom-designed genetic algorithm, GA SARAS, into the AEG module, which efficiently synthesizes excitation coefficients that match complex, multi-objective, two-way radiation masks specifically generated for each beam. The generation of these masks involves a sophisticated process in which they are tailored to meet multiple simultaneous constraints of both the antenna and the SAR system. These constraints include gain uniformity, sidelobe level suppression, and pointing accuracy and cover system-level requirements such as the NESZ, gain, AASR, SLL, and RASR and a 3 dB beamwidth (Section 3.4).

Another major contribution is the design of a dedicated cost function that guides the optimization process in alignment with the multi-objective radiation masks. This cost function is built by dividing the angular domain into distinct zones, assigning each a weight proportional to its relevance to the mission objectives. This zoned-weighting structure directs the optimization toward solutions that meet all of the required design criteria, as elaborated further in Section 3.5.

Finally, in contrast to many previous studies that are based on idealized assumptions or entirely simulated environments, the proposed framework has been validated using real experimental data from the PAZ satellite. This validation includes actual subarray measurements, the detailed physical geometry of the antenna array, and the operational status of the transmit–receive modules (TRMs), thereby significantly reducing the gap between simulations and real-world implementation.

In order to describe the new features of this article, this article is organized as follows: Section 2 describes the SARAS model, model objectives, graphical interface, and APG and APE modules. Section 3 describes the genetic algorithm designed for this model, including the choice of PRF procedure and the generation of a suitable mask and cost function, which are the key parts for this model to work properly. Section 4 describes the validation process using experimental data used in the PAZ satellite currently in orbit and shows the results for PAZ and ECO (Figure 1), comparing the genetic algorithm presented with other optimization algorithms in the literature and demonstrating the benefits of the design presented, and finally, Section 5 shows the conclusions.

## 2. The Antenna Model Approach

### 2.1. The Purpose of the Antenna Model

In space SAR calibration, a reliable antenna model is crucial for system accuracy. It provides a mathematical representation that dynamically accounts for variations in the active array using internal calibration data. This model reconstructs radiation patterns from ground-measured parameters and excitation coefficients, reducing the number of beams that require in-flight measurement and allowing their refinement using flight data. The model is based on measurements of a single subarray in free space [15,16,17,18,19].

This section delineates the implementation of a phased-array antenna model for Synthetic Aperture Radar Systems (SARAS) in the MATLAB R2024a environment developed by the INTA’s Space Based SAR Systems and Calibration Group. The utilization of this antenna model is of particular pertinence in the context of the PAZ and PRECURSOR ECO Spanish Earth observation missions. The antenna model incorporates numerous functions, which are divided into two primary modules: the first module, antenna pattern generation (APG), and the second, antenna excitation generation (AEG), work together to produce optimized radiation patterns tailored to specific multi-objective masks, using a custom-designed genetic algorithm (GA) [19].

**The antenna pattern generator (APG)** is employed to generate the reference antenna patterns that are required in the ground segment for pattern compensation in the SAR processor and for radar parameter generation. This pattern generation module was validated using PAZ satellite data.

An analysis of the radiation patterns in the phased-array antenna is performed by combining the measured responses of the unitary elements (element-level patterns) and the physical layout and geometry of the antenna array, including mechanical errors and the operational status of the transmit–receive modules (TRMs), including any identified failures. This information is used to compute the overall array factor and synthesize the resulting radiation pattern, accurately reflecting the real performance of the antenna in its current configuration. All of this information is stored and updated in XML input files so that it can be used for any space mission. The origin of the antenna reference system is at the geometric center of the antenna’s radiating plane, and the main beam nominal pointing direction is boresight, when the antenna beam points towards the z-axis. This direction is at θB = 33.8° from the nadir (Figure 2).

The antenna angles θ and φ are related to x, y, z by Equation (1)(1)xyz=r·sinθ·cos∅sinθ·sin∅cos∅with 0≤ϑ≤2π0≤∅≤π0≤r≤∞

A phased-array antenna is composed of a set of N × M radiating elements, which are arranged into a regular pattern. The relative phases and amplitudes of the currents that feed these elements result in a specific radiation pattern. The pattern ${\vec{E}}_{A}\left(\theta ,\emptyset \right)$ is computed as a superposition of the fields radiated by the N × M (elevation × azimuth) subarray patterns $\left|{\vec{E}}_{e}\left(\theta ,\emptyset \right)\right|$ weighted by the complex excitation coefficients given by the TRM setting $\left|F_{A}\left(\theta ,\emptyset \right)\right|$. Subindexes m and n specify a subarray by column and row, respectively, in the antenna array. The subarray patterns are considered to be embedded into the total antenna so that all relevant effects are considered in the simple formula [18]:(2)$$\left|{\vec{E}}_{A}\left(\theta ,\emptyset \right)\right|=\left|{\vec{E}}_{e}\left(\theta ,\emptyset \right)\right|\cdot \left|F_{A}\left(\theta ,\emptyset \right)\right|$$
where ${\vec{E}}_{A}\left(\theta ,\emptyset \right)=\sum_{i}{\vec{E}}_{i}\left(\theta ,\emptyset \right)={\vec{E}}_{e}\left(\theta ,\emptyset \right)\cdot \sum_{i=1}^{N}A_{i}\cdot e^{jK_{0}\vec{r}\vec{r_{i}}}$ and

$F_{A}\left(\theta ,\emptyset \right)=\sum_{i=1}^{N}A_{i}\cdot e^{jK_{0}\vec{r}\vec{r_{i}}}$ with $\bar{r}{\bar{r}}_{i}=md_{x}sin\theta cosɸ+nd_{y}sin\theta sinɸk_{0}=\frac{2\cdot \pi }{\lambda }$with $A_{i}=a_{m}e^{jm\alpha_{x}}\;.\;a_{n}e^{jn\alpha_{y}}$

Each radiating element ${\bar{E}}_{e}(\theta ,ɸ)$ is fed by a complex feed coefficient (amplitude and phase), $A_{i},\;$and each radiating element is placed in a specific position in space (linear, rectangular, circular, etc.) defined by the offset from the origin, $e^{jk_{0}\bar{r}{\bar{r}}_{i}}$ (Figure 3).

Multiplication of the last two terms gives the array factor ${\bar{F}}_{A}(\theta ,ɸ)$, which contains information about the geometry (the displacement from the origin) and the excitation law of the radiation (the amplitude and phase of the complex feed coefficients). Finally, the radiation pattern of an array is the product of the radiation pattern of a single radiating element and the array factor [18,19,20,21]. In practice, the generation of radiation patterns in a phased-array antenna, accounting for both TR module deviations and mechanical deformations, involves a rigorous, multi-tiered modeling approach [17,20].

With regards to TRM deviations, the excitation coefficients are then corrected to account for TRM-level impairments. Amplitude and phase deviations are modeled by applying perturbations to each Ai (2), such that the effective excitation becomes (3)(3)$$A_{i}^{eff}=(a_{m}+{∆}_{ami}{)e}^{j(m\alpha_{x}+{∆}_{\alpha_{xmi}})}\;.(\;a_{n}+{∆}_{ani})e^{j(n\alpha_{y}+{∆}_{\alpha_{yni}})}$$
where ${∆}_{ami}$, ${∆}_{ani},\;{∆}_{\alpha_{xmi}},\;and\;{∆}_{\alpha_{yni}}$ represent the amplitude and phase errors introduced by TRM imperfections and are read from a TRM status file in XML format.

Mechanical deformations in the antenna system can be simulated at three hierarchical levels. At the subarray level, thermal variations cause parabolic z-axis displacements in the annular slots within each subarray, with user-defined maximum deviations. At the panel level, structural inaccuracies introduce x-, y-, and z-axis displacements and roll, pitch, and yaw rotations in the subarrays, which are simulated by modifying their positions and rotating the measured patterns accordingly; typical z-axis deviations are around ±0.5 mm. At the antenna level, entire panels experience z-axis displacements and rotations (up to ±1 mm and 0.2° in pitch and yaw), which are applied globally to the antenna pattern. Mechanical deformations are then incorporated by adjusting the spatial positions of the subarray elements $\bar{r}{\bar{r}}_{i}’$ = (xi, yi, zi) to the ideal position vector of the i-th element (2). The deformed position becomes(4)$$\bar{r}{\bar{r}}_{i}’=T_{panel\;}\cdot T_{subarray\;}\cdot (\bar{r}{\bar{r}}_{i}+∆\bar{r}{\bar{r}}_{i})$$
where $∆\bar{r}{\bar{r}}_{i}$ models the thermal or structural displacement at the subarray level, and $T_{panel\;}and\;T_{subarray\;}$ are transformation matrices encoding the rotations and translations due to mounting misalignments at the subarray and panel levels, respectively (including roll, pitch, yaw, and z-axis shifts).

The directivity for the TX and RX co-polar generated patterns can be accurately computed by integrating the total field radiated by the subarray using Equation (5):(5)$$D_{cop}=4\cdot \pi \frac{{\left|E_{NxM}^{cop}(\theta ,\emptyset )\right|}^{2}}{\int_{0}^{2\pi }\int_{0}^{\pi }\left[{\left|E_{NxM}^{cop}(\theta ,\emptyset )\right|}^{2}+{\left|E_{NxM}^{cross}(\theta ,\emptyset )\right|}^{2}\right]\sin\theta d\theta d\emptyset }$$

This complete model ensures that the resulting pattern accurately reflects both electronic and mechanical non-idealities, enabling more robust predictions and compensation for real-world SAR mission performance.

**The antenna excitation generator (AEG):** The overarching objective of the AEG module is to generate all possible antenna beams that are required to operate and calibrate a SAR system. The generation of a set of complex excitations corresponding to the required beam from different satellite operating swaths is dependent on the antenna model, which is required to do so. This is achieved by starting from several radiometric parameters that have been defined in advance. The importance of optimization in this context is crucial because it facilitates the re-computation of excitations in the event of failures or drifts in the transmission and reception modules (TRMs), which in turn feed the antennas forming the array (Figure 4). To carry out this optimization effectively and reliably, AEG develops an ad hoc genetic algorithm, so that the algorithm compares the computed result against a previously obtained mask to meet the radar and antenna parameters in a more efficient and accurate way than the other methods found in the literature [9,10,11,12]. From this comparison, fitness values are obtained to evaluate how good the solution is and to continue iterating or reach a stop condition.

The main functions of the AEG module are as follows:

1. Configure the pashed array with user-selected XML format input files containing the subarray measurements, antenna geometry, TRM status, and antenna geometry. 2. Select the beams to be synthesized and the number of iterations. 3. Either allow a mask already created from the file to be read or generate it again. In this case, the user interface displays a series of parameters to be filled in, such as the PRF, Doppler centroid, nadir area, etc. 4. Optimize the best diagram that complies with the selected or created mask, showing the evolution of the algorithm by means of output graphs during the process. 5. Save the generated amplitude and phase coefficients into XML files which can be read by the APG.

### 2.2. The User Interface of the SARAS Antenna Model

The SARAS model is intended for the design of the antenna patterns for all planned SAR antenna operating modes. This is carried out with the aim of achieving the maximum performance in the collection of data on an operational basis. For this reason, the GUI distinguishes between the synthesis (AEG) and analysis procedures (APG), following an array description definition procedure (Section 2.1). The toolbar in the main antenna model window, shown in (Figure 5), is the link to the windows that perform the previously mentioned functionalities (1. Array description, 2. Errors, 3. Analysis, 4. Synthesis, and 5. Preprocessing and status) in Figure 6. The Antenna Modeler is implemented using MATLAB R2024a and is designed to run on personal computers.

As illustrated in Figure 5 (left), both the APG and AEG modules can be selected within the primary interface. Furthermore, both modules utilize a consistent graphical interface. When the APG module is selected, a pattern analysis is performed, and when the AEG is selected, the patterns are synthesized according to the input requirements. Upon the termination of the simulation, both the mask and the excitation coefficients are documented in XML files and stored in the designated output directory, as specified in the graphical interface. The graphs are stored in the Portable Network Graphics (PNG) format, and the information displayed on the About screen is saved in a log file. Finally, the result summary tab displays a summary of the antenna parameters utilized in the simulation (Figure 5 right). This includes the beams for synthesis, in addition to the final optimized radiation pattern for each beam and the excitation coefficients that generate each synthesized pattern.

## 3. The Excitation Coefficient Optimization Tool (AEG) Based on a GA

### 3.1. The Optimization Framework for Phased Arrays in SAR Systems

As mentioned above, synthetic aperture radar (SAR) has been extensively utilized in remote sensing due to its capacity to acquire high-resolution images irrespective of the weather conditions, even during diurnal and nocturnal periods. In order to obtain SAR images of the area of interest (AOI), contemporary SAR systems employ an active phased-array antenna to electronically generate multiple beams. The beams must meet specific objectives, which are defined not only by conventional antenna parameters (e.g., gain, steering, cross-polar levels, null positions) but also by the SAR parameters (e.g., range ambiguity level, swath flatness). In scenarios where the pattern shape is constrained by numerous practical factors, the employment of optimization techniques is imperative. The objective of optimizing this particular antenna model is to achieve a trade-off that considers the full set of constraints [17,18,19,20,21]. A variety of optimization techniques have been employed in this domain, each of which possesses distinct strengths and limitations (Figure 7).

Classic iterative optimization methods such as gradient descent, quasi-Newton [22], Least Squares Methods (LSMs) [23,24], and convex optimization techniques are well suited to problems that are smooth, continuous, and convex. These methods are deterministic and rely on analytical gradients to converge quickly and efficiently toward a solution. However, they often encounter significant limitations when dealing with non-linear, non-convex, or discrete problems—conditions commonly found in antenna array synthesis and electromagnetic design. For these scenarios, evolutionary algorithms like genetic algorithms (GAs) [25,26,27,28,29], Particle Swarm Optimization (PSO) [30,31,32,33], and Differential Evolution (DE) [34] offer a more suitable alternative. These algorithms are stochastic, population-based, and inspired by natural processes, which makes them inherently robust and flexible [25,26,27,28,29,30,31,32,33,34,35,36,37].

Among these, GAs have shown particular effectiveness for multi-objective optimization problems, thanks to their ability to manage diverse constraints through custom cost functions and mask-based formulations. They also benefit from genetic diversity and operators such as crossover and mutation, which facilitate global exploration and reduce the risk of convergence to local minima. Compared to DE and PSO, a GA is more adaptable for phased-array synthesis in spaceborne SAR systems, particularly when faulty TRMs, measured radiation patterns, and structural constraints are involved [25,26,27,28,29,30,31,32,33,34,35].

In this study, an ad hoc version of a GA has been developed specifically to enhance the performance in this demanding multi-objective environment. This customized GA improves the convergence efficiency and ensures better alignment with the multi-objective mask, outperforming the conventional algorithms in both robustness and solution quality. The complexity and slower convergence of a GA are acceptable trade-offs given the mission’s strict image quality requirements and the necessity to avoid incorrect solutions due to premature local minima convergence.

### 3.2. The GA Implemented by INTA

In the context of generic genetic algorithms (GAs), populations undergo evolutionary changes across successive generations in accordance with the principles of natural selection and survival of the fittest, as postulated by Darwin. Each solution is assigned a value or score, which is determined by the goodness of that solution. The greater the adaptation of an individual to a given problem, the higher the probability of that individual being selected to reproduce by crossing its genetic material with another individual who has been similarly selected. It has been demonstrated that this crossover will produce new offspring which exhibit some of the characteristics of their parents. The degree of adaptation exhibited by an individual is directly correlated with the probability of that individual being selected for breeding. Consequently, this also determines the likelihood of the genetic material of that individual being propagated across successive generations. In this manner, a novel population of potential solutions is engendered, which supplants the preceding one and corroborates the intriguing property that it comprises a greater proportion of advantageous characteristics in comparison to the previous population [25,26,27,28,29].

The GA implemented by INTA [19] operates iteratively, during a defined number of iterations, on a group of experimental solutions in parallel, termed the population. For each beam selected in the GUI to be synthesized, the initial population, comprising N individuals, is selected to be entirely random. Each solution within this population, designated as an individual, exhibits a vector of entries known as genes (complex excitation coefficients: module and phase for reception and only phase for transmission (Figure 8). The algorithm utilizes the XML input files containing the antenna configuration and geometry and TRM drifts, in addition to the measured anechoic chamber patterns embedded within the unit elements. Furthermore, the given two-way mask, which is also in XML format, must be read. In order to synthesize the two-way pattern, a comparison of this mask with the aforementioned pattern is necessary. It is also possible to create this mask, with the proviso that the radar parameters are introduced via commands in the SARAS user interface (see Section 3.4). The two-way radiation patterns of the current population (parents) are configured using the phase and amplitude values for the reception and transmission patterns applied to each element of the array, the array geometry, and the radiation patterns of the individual elements. Furthermore, the term “parent” is used to denote a member of the current generation, while “child” is used to denote a member of the next generation (see Figure 8). In each iteration, the individuals (radiation patterns) are evaluated and ordered according to a fitness function defined by INTA as a comparison between the two-way antenna gain pattern and a multi-target mask. This results in a table of ordered fitness values.

Each fitness value is a positive number assigned to an individual to facilitate comparisons with other individuals in a population, thereby representing a measure of the goodness of each solution. Subsequent to the ranking of the current population according to its fitness, the next population should be populated through the reproductive cycle (Figure 9).

The operators, modified ad hoc for the designed genetic algorithm, are now starting to be applied: elitism is a mechanism for filling the next generation, and it is defined as the direct transition of the best individuals in a given generation to the next, without alterations to their genes (see Figure 9). The subsequent stage of the process is to perform partial hypermutation. In this process, with a 50% probability, a mutation will be introduced into the new generation, thereby introducing new ‘genetic material’. In the ad hoc algorithm, this process is defined by the random generation of a new individual (amplitude and phase) in 50% of cases, which subsequently occupies the final position in the newly created population (Figure 8). The subsequent phase is the establishment of the kindergarten, which is defined as the process of procreation. With regard to the reproduction cycle implemented by INTA, the individuals with the highest fitness levels (i.e., the parents) are selected by means of a sliding window technique in order to combine their genes and create a new solution (i.e., the child) for the next generation by means of a crossover operator (i.e., a neighborhood-based technique) with a crossover probability. The final mechanism for the propagation of the ensuing generation is known as ‘mutation BGA’ [36]. This process, modified by INTA, entails the random introduction of novel ‘genetic material’ into the population through the alteration of genes in pre-existing individuals. It is evident that under the optimal conditions, a well-designed genetic algorithm will result in the population converging towards a solution that can be considered optimal. This solution pertains to the complex excitation coefficients required for each phased-array element, and it has the capacity to satisfy the specified radar requirements at the antenna level.

In genetic algorithms, careful tuning of parameters such as the mutation rate, crossover probability, and elitism is essential to ensure convergence within a reasonable number of iterations. Moreover, the construction of a suitable multi-objective mask and the formulation of an effective cost function to guide the optimization process are critical for achieving efficient convergence and high-quality results. The success of the GA largely depends on these design choices. Therefore, the following subsections are dedicated to detailing the development of the multi-objective mask and the corresponding cost function.

### 3.3. Strategies Implemented for Optimization

The main objective of the “SARAS” antenna model is to analyze and synthesize beams defined for a space-based SAR mission. The SAR mission will determine the radar and antenna parameters that must be met, the AASR, RASR, NESZ, SLL, BW-3dB, gain, etc., and for this, the phased array must be specifically designed and optimized.

The AASR (azimuth-ambiguity-to-signal ratio) measures the level of azimuthal interference caused by ambiguities with respect to the desired signal. This parameter depends mainly on the PRF, which must be high enough to avoid aliasing in the Doppler domain, as well as on the azimuth beamwidth and the radiation pattern of the antenna, especially the level of its side lobes. In addition, factors such as the satellite speed, observation geometry, and SAR processing techniques also play a role, as they affect the spectral distribution and the ability to filter out ambiguities. Careful design of these elements reduces the AASR and improves SAR image quality.

The RASR (range-ambiguity-to-signal ratio) quantifies the level of range interference caused by ambiguous echoes from other pulses, relative to the main signal. A low RASR is essential to ensure high radiometric quality and avoid image contamination. The RASR mainly depends on the PRF, the target range, and the antenna’s elevation radiation pattern [6,7,8,9,37].

The NESZ (noise-equivalent sigma zero) is a measure of radar sensitivity and represents the minimum level of terrain reflectivity that can be detected above system noise. A low NESZ value indicates higher sensitivity and a better ability to detect faint targets. The NESZ depends on several factors, including the antenna gain, beamwidth in elevation and azimuth, and transmit and receive efficiency. In particular, antennas with higher gain and narrower beams concentrate the energy better, reducing the NESZ. Operational parameters such as the target distance (range), operating frequency, and transmitted power also play a role [6,7,8,9,10,11,12,13,37].

As established in the previous definitions, the pulse repetition frequency (PRF) is a critical parameter that directly affects both the azimuth-ambiguity-to-signal ratio (AASR) and the range-ambiguity-to-signal ratio (RASR). Therefore, the optimization process begins with the careful selection of a suitable PRF that balances these competing requirements while satisfying the mission constraints.

However, selecting an appropriate PRF also requires prior knowledge of the swath width, as these two factors are closely interdependent. Therefore, while the goal is to define a PRF that balances the mission requirements and satisfies the threshold values for the AASR and RASR (as detailed in Table 1), this cannot be carried out without first analyzing the swath configuration—addressed in the following step.

The following step focuses on achieving complete coverage of the mission-defined observation area, which is divided into multiple beams, each with a swath width typically ranging between 20 km and 30 km. For each swath, the corresponding beam steering angles are calculated to ensure that the radiometric and image quality requirements are met.

To support this analysis, it is necessary to evaluate the range of valid PRFs for each beam based on acceptable ambiguity levels (Figure 10a). Since the swath is constrained by the time available to receive echoes between transmitted pulses, a high PRF shortens the reception window and limits the observable range—resulting in a narrower swath. Conversely, a lower PRF allows for longer reception times and thus wider swaths but may increase ambiguities. Therefore, the PRF must be carefully selected to minimize both the azimuth (AASR) and range (RASR) ambiguities. Furthermore, because the PRF requirements vary along the satellite’s orbit, the analysis also considers the orbital segments where each PRF remains valid (Figure 10b).

Bearing this in mind, the associated diamond diagram is generated for the proposed swaths between 20 and 30 km, as shown in Figure 10a. The diamond diagrams (PRFs vs. angles of incidence) are analyzed to identify PRF values that are within the unambiguous zones for the azimuth and range and which comply with this swath. In the diamond diagram, the valid region for PRF selection is defined as the area between two critical curves. The lower bound (PRFmin) ensures that azimuth ambiguities are avoided by preventing Doppler aliasing. The upper bound, PRFmax (6), ensures that range ambiguities are avoided by allowing sufficient time for echo reception before the next pulse is transmitted. The region between these two bounds represents the valid PRF range that satisfies both ambiguity constraints, and only PRF values within this region can be considered viable for system design.(6)$${PRF}_{min}=\frac{2\cdot vsat\cdot sin(\theta_{i})}{\lambda }\;{PRF}_{max}=\frac{c}{2\cdot Rmax}\;c:\;speed\;of\;light$$(7)$$AASR=\frac{\sum_{m=-\infty }^{m=\infty }\int_{-B/2}^{B/2}{PMLL}^{2}(f+m\cdot PRF)df}{\int_{-B/2}^{B/2}{PMLL}^{2}(f)df}$$

Therefore, for each proposed swath, the diamond diagram represents several valid PRFs for each beam (angular ranges). To choose the best PRF associated with the best swath, an optimization process is carried out based on a weighted evaluation of the variables involved. The highest weight is assigned to the PRF values with the lowest magnitude, prioritizing lower PRFs. The next weight is given to PRF values that yield a better AASR (7) and RASR performance. Following this, a lower weight is assigned to PRFs that appear more frequently along the orbit, favoring temporal consistency. Finally, the smallest weight is assigned to PRFs that exhibit minimal jumps or discontinuities in their distribution along the orbit, contributing to operational stability and ease of implementation. At the end, for each beam, the swath, pointing angles, and associated PRFs are chosen and stored directly in the configuration table of the SARAS antenna model.

As specified in Table 2, the elevation steering range of PRECURSOR-ECO (−19.54° to 13.84°) is divided into 25 swathes, taking into account θB = 33.8°. The parameters chosen in the prior analysis have been systematically examined to fulfil the mission objectives described in Table 1, with their specific values presented in Table 2.

### 3.4. Mask Definition

At this point, it is necessary to calculate a multi-target mask that makes the synthesized pattern meet these requirements. Range ambiguities are caused by the preceding and succeeding pulse echoes reaching the SAR antenna, together with the desired return signal (Figure 11). The primary consequence of these range ambiguities is the manifestation of “ghost targets” or false signals in the SAR image. These artifacts have the potential to be misinterpreted as actual terrain features, thereby complicating the process of image interpretation and analysis. Furthermore, range ambiguities can have a detrimental effect on the accuracy of Doppler velocity estimation and target detection, particularly in high-resolution applications [37]. In consideration of the aforementioned factors, the generation of a beam pattern with reduced sidelobe levels across all angular orientations presents a considerable challenge. The reduction in sidelobes at specific angles tends to result in an increase at other angles, thereby complicating the desired pattern. Consequently, it has been demonstrated that beam patterns exhibiting low sidelobes exclusively in the direction of significant ambiguous echoes are more effective for suppression. The main lobe beam pattern of the SAR antenna is associated with the noise-equivalent sigma zero (NESZ) and the swath width, whereas the sidelobe beam pattern is related to the azimuth-ambiguity-to-signal ratio (AASR) and the range-ambiguity-to-signal ratio (RASR). As the pulse repetition frequency (PRF) increases, the azimuth ambiguity in the along-track direction decreases, whereas the range ambiguity in the cross-track direction increases [6,7,8,9,10,37]. The purpose of this particular focus has been to enhance the RASR performance through the optimization of elevation beam patterns under relatively high-PRF conditions, with the objective of ensuring the security of the AASR and NESZ performance [11,12,13].

Given the results obtained after the PRF analysis in Section 3.3, it becomes essential to define a two-way elevation beam pattern mask in which the sidelobe levels are inversely proportional to the estimated power levels of range-ambiguous echoes. The integrated range-ambiguity-to-signal ratio (RASR) is formally defined as the ratio of the total power of all undesired signal components, originating from both preceding and subsequent pulse echoes, within the receiving (or recording) window to the integrated power of the desired signal return (8). First of all, the relative power levels of the ambiguous echoes are computed based on a backscattering coefficient model, the satellite altitude, the target incidence angle, and the pulse repetition frequency (PRF) in order to obtain the ambiguous regions, as illustrated in Figure 11.(8)$${RASR}_{integrated}=\frac{\sum_{j=1}^{N}S_{aj}}{\sum_{j=1}^{N}S_{j}}$$
where Saj and Sj are the range ambiguities and the desired return signal power in the j-th time interval of the data recording window, respectively. N is the total number of time intervals (9, 10):(9)$$S_{a}=\sum_{k=-n}^{k=n}\frac{\sigma_{0}(\theta_{k})\cdot D_{2way}(\theta_{k})}{R_{k}^{3}(\theta_{k})\cdot sin(\theta_{k})}$$for k ≠ 0(10)$$S=\frac{\sigma_{0}(\theta_{0})\cdot D_{2way}(\theta_{0})}{R_{0}^{3}(\theta_{0})\cdot sin(\theta_{0})}$$
where θ_0_ is the incidence angle at a certain point in the desired swath, and θ_k_(k = 1, 2, …, n) is the incidence angle at the range-ambiguous location. The subscript k represents the preceding and succeeding pulse echoes and is also used as an index to indicate one of several ambiguous regions. The following three notations, σ0, R, and D_2way_, are the backscattering coefficient, the slant range, and the two-way directivity in the range direction, respectively, all of which depend on the incidence angle. As shown in Figure 11b, all regions where the adjacent slant range distance differs by c/2PRF [m] become range-ambiguous regions. These ambiguous regions are shown in Figure 12 for a specific beam of the ECO satellite defined with a pointing angle of −17.98° (strip_001), a PRF of 3650, a Doppler centroid of 44,000, and a processing bandwidth of 2765. The antenna has dimensions of 0.75 m × 3 m and a rectangular geometry of 32 × 12 elements. It should be noted that the satellite is tilted by 33.8°; therefore, to calculate the ambiguous regions, a correction of 33.8° must be applied to the look angle, resulting in a steering angle of 15.81°.

Once the ambiguity areas have been identified (Figure 12a), both the inner and outer masks are calculated (Figure 12b) to ensure that the sidelobes are low in the ambiguous regions while maintaining an adequate NESZ level (Table 1).

To create an optimum inner beam pattern mask, we performed the following tasks. First, we determined the two-way gain of the main lobe (the maximum value of the inner mask PMLL) based on the required minimum NESZ (11), where R is the slant range, PN is the system noise power, Ltot is the total loss of the SAR system, v_sat_ is the satellite height, PRF is the pulse repetition frequency, and $\tau_{chirp}$ is the pulse width.(11)$$PMLL=10log(\frac{4\cdot {4\pi }^{3}\cdot v_{sat}{{\cdot L}_{tot}\cdot PN\cdot sin(\theta_{i})\cdot R}^{3}}{\lambda^{3}\cdot c\cdot NESZ\cdot \tau_{chirp}\cdot P_{tx}\cdot PRF})$$

After finding the antenna gain (inner mask), the main lobe beamwidth, which decides the azimuth resolution, needs to be determined. The relation between the azimuth resolution and the main lobe beamwidth at the reference level of 3 dB is given in (12). The RA is calculated according to the flatness of the main lobe, and the flatter the main lobe, the better the RA. The angles that define the position of the inner mask are specified in an XML file, as look_angle_max and look_angle_min, calculated from the swath for each specified beam.(12)$${BW}_{-3dB}=\frac{\lambda }{2\cdot Azimuth\_resolution}$$

For the design of the beam two-way gain pattern mask (Figure 13), the incidence angles (θ) need to be converted into look angles (α). The sidelobe level of the normalized directivity pattern mask for each look angle (αk), represented with blue lines, corresponding to the ambiguous location and can be derived as (13)(13)$$D_{2wayNLL}\left(\alpha_{k}\right)=\frac{RASRreq}{2n}\cdot \frac{\frac{\sigma_{0}(\alpha_{0})}{R_{0}^{3}(\alpha_{0})\cdot sin(\alpha_{0})}}{\frac{\sigma_{0}(\alpha_{k})}{R_{k}^{3}(\alpha_{k})\cdot sin(\alpha_{k})}}$$

### 3.5. The Cost Function or Fitness

To evaluate candidate solutions, the SARAS GA uses a fitness function ranging from 0 (poor) to 1 (optimal). This function (14) compares the synthesized radiation pattern against a multi-objective mask divided into angular zones (e.g., main beam, ripple, roll-off, NRLL), each weighted according to mission priorities like the NESZ, AASR, or gain. Errors between the pattern and the mask’s inner and outer limits are calculated per zone and combined using weighted coefficients (a, b, c, d). This guides the optimization toward solutions that meet all of the constraints across the full angular range (Figure 14).

To assess the degree of adaptation, a comparison is made between the outer and inner masks and the radiation patterns synthesized for each member of the population at each iteration of the algorithm. Four types of error are defined for this purpose: **eRipple** refers to the error that occurs when the synthesized beam extends beyond the 3 dB beamwidth, i.e., into the area shaded in purple (15); **eNRLL** (the error in the normalized relative sidelobe level), which is divided into right and left errors with respect to the steering angle. Each of the right and left parts is divided further into three zones (blue, orange and yellow areas), which are weighted according to how much each zone affects the ambiguities (16,17); **eMainBeam** (the error in the main beam), which occurs when the synthesized beam falls below the inner mask or exceeds it due to excessive ripple (18,19), is shaded in red and light blue. Finally, the out-of-slope error (**eSlides**) is calculated as the sum of the points that do not follow the slope, both left and right, relative to the points within the green area (20,21). By calculating all of the errors and applying the appropriate weighting factors (a, b, c, and d, which indicate the relative importance of side lobe errors versus main beam errors), the fitness value is determined.(14)$$fitness=\frac{1}{1+a\cdot eNRLL+b\cdot eMainBeam+c\cdot eRipple+d\cdot eSlides}$$(15)$$eRipple=\frac{\sum_{i=1}^{n}\left(Pi-OMi\right)>0}{n}$$(16)$${eNRLL}_{left}=\frac{\sum_{i=1}^{nl}Pi-OMi>0}{nl}+\frac{\sum_{i=1}^{ml}\left(Pi-OMi\right)>0}{ml}+\frac{\sum_{i=1}^{ll}\left(Pi-OMi\right)>0}{ll}$$(17)$${eNRLL}_{right}=\frac{\sum_{i=1}^{nr}\left(Pi-OMi\right)>0}{nr}+\frac{\sum_{i=1}^{mr}\left(Pi-OMi\right)>0}{mr}+\frac{\sum_{i=1}^{lr}\left(Pi-OMi\right)>0}{lr}$$(18)$${eMainBeam}_{int}=\frac{\sum_{i=1}^{r}\left(OMi-Pi\right)>0}{r}$$(19)$${eMainBeam}_{ext}=\frac{\sum_{i=1}^{p}\left(Pi-IMi\right)>0}{p}$$(20)$${eSlide}_{left}=\frac{\sum_{i=1}^{sl}\left(out\;of\;left\;slope\;points\right)}{sl}$$(21)$${eSlide}_{right}=\frac{\sum_{i=1}^{sr}\left(out\;of\;right\;slope\;points\right)}{sr}$$
with (Pi-OMi)=$\left({Pattern}_{i}-{outter_{Mask}}_{i}\right)$, (OMi-Pi)=$\left({Outter_{Mask}}_{i}-{Pattern}_{i}\right)$, (Pi-IMi)=$\left({Pattern}_{i}-{Inner}_{maski}\right)$.

The weights a, b, c, and d were selected after extensive testing to ensure rapid convergence (under 150 iterations) and fitness values above 0.7, indicating strong mask compliance within the −50° to 50° range.

## 4. Validation of the SARAS Antenna Model with Experimental Data from the PAZ Satellite

This section shows results that validate the model—on the one hand, results validating the APG module of the antenna model, based on measurements of a leaf (one third of the PAZ satellite antenna currently in use), the antenna geometry, and the subarrays. On the other hand, the results of the AEG module will be shown, which synthetized diagrams for all of the beams specified by parameters or by file, both for the PAZ satellite and the PRECURSOS-ECO satellite, currently under development by INTA. The rest of the specified parameters, such as the RASR, NESZ, and gain for each of the beams currently considered for PRECURSOR-ECO, will also be shown.

### 4.1. APG Results and Validation

To validate the APG model, the conformity between the modeled reference antenna patterns and the corresponding on-ground measured antenna patterns (leaf) has been verified.

This section uses the PAZ antenna’s subarray patterns measured in the anechoic chamber, as well as their geometrical characterization within the antenna and the state of the TRMS at the time of averaging. The ground measurements of the leaf (one third of the PAZ antenna) are compared with the pattern calculated through an analysis for the leaf, and the model can be validated as the gain error is negligible (below the measurement error of 0.4 dBi), and the side lobes, despite being measured with low precision, present levels in accordance with the model simulated using SARAS (Figure 15).

### 4.2. AEG Results, Beam Synthesis, Comparison, and Performance Evaluation

This section describes the pattern synthesis results produced by the AEG module and the creation of a fitness function and masks that fulfil the requirements of the antenna and radar parameters. SARAS has the peculiarity of having been designed for pattern synthesis of any space phased-array antenna, offering users the ability to modify the antenna geometry and configuration, as well as perform unit element measurements from files or manually (Figure 6). It has therefore been used with the PAZ satellite to compare existing results and validate the model, as well as with the new PRECURSOR-ECO satellite.

The PRECURSOR-ECO antenna with a size of 0.75 × 3m will consist of 32 elements in elevation and 12 in azimuth, divided into six panels and grouped into three leafs. One fixed panel and the two end panels will be deployable after launch. Each unit element will be a carbon fiber slotted waveguide antenna that has already been designed, manufactured, and measured (Figure 16). This unit element has been designed to ensure that the complete antenna complies with an AASR below −17dB (Figure 17b), complying with an azimuth mask not only at the center frequency but over the entire band, in order to fulfil the AASR (Figure 17).

With regard to the elevation cuts, the elevation diagram (Figure 18) displays the two-way diagrams synthesized using SARAS for all of the beams defined in the mission (Table 2). As illustrated in Figure 19, each of the beams under consideration complies with its mask. As demonstrated in the accompanying diagrams, beams in closer proximity to the nadir necessitate beam flattening and tapering in order to become less directive than the beams pointing away from the nadir, which exhibit a more brush-like character. The beam near the nadir is typically wider in order to cover a greater area in a more uniform manner, and the nadir beam does not face severe elevation ambiguities. In contrast, the extreme beams (those at the edges of the swath) are usually more directional in order to avoid gain loss and ambiguities and to comply with the elevation ambiguity mask. Furthermore, beams situated at a considerable distance from the nadir are more susceptible to elevation ambiguities, as their secondary lobes may intersect with sensitive areas of the terrain. To mitigate the level of these ambiguities, these beams are successfully designed to be narrower (higher directivity).

After running the SARAS AEG antenna model for all of the beams defined for PRECURSOR-ECO, four examples of synthesized two-way antenna elevation patterns are presented (Figure 19). These patterns are optimized to comply with a mask previously calculated using radar parameters, as described in Section 3.4. Equally, for these four examples, the two-way gain diagrams are synthesized using different analytical and optimization algorithms such as Taylor [38], Chebyshev [39], an LSM [23,24], DE [34], PSO [30,31,32,33], a generic GA [25,26,27,28,29], and GA SARAS and compared with each other. These analyses are carried out under the same conditions in terms of the number of iterations (between 150 and 1000) and optimization mask.

Although the Chebyshev and Taylor methods are computationally efficient and capable of producing low sidelobe levels (SLLs), they lack the flexibility to broaden the beamwidth in beams near the swath edge (Figure 19 (a): Strip_001; (b): Strip_004; (c): Strip_010; (d): Strip_025 (a)). This limitation stems from their poor performance in complex multivariable optimization problems like the one addressed in this study [38,39]. On the other hand, metaheuristic algorithms such as PSO, the standard GA, and DE are able to follow the target mask reasonably well for near-nadir beams, where the ambiguity levels are relatively low. However, their performance significantly degrades as the beam steering angle increases (Figure 19 (a): Strip_001; (b): Strip_004; (c): Strip_010; (d): Strip_025 (b,c)) [25,26,27,28,29,30,31,32,33,34]. These methods tend to either reduce the SLL or improve the gain in the main lobe region, but within the same number of iterations used by the SARAS GA, they fail to fulfill the entire multi-objective mask. While these algorithms are generally faster, they are not well suite to problems involving a high number of time-dependent variables, such as those encountered in phased-array optimization.

Once synthesis of all of the beams defined for ECO-PRECURSOR has been completed, the performance of these beams is analyzed to verify that the tool is able to synthesize diagrams according to a mask (Figure 20), thus fulfilling the necessary requirements for each beam (Table 1 and Table 2).

Figure 20a shows the AASR values calculated with the beams synthetized for ECO-PRECURSOR with the SARAS AEG. The thick red line shows the minimum allowable threshold (−17 dB), while the thick yellow line shows the target to be reached. It can be seen that all of the beams meet the threshold.

Figure 20b shows the NESZ value for each beam, with the thick yellow line being the threshold defined in the requirement tables. In the same way, it can be seen that the masks calculated for elevation are adequate, as the synthesized beams comply with the NESZ (green dotted line), not only in the center of the beam but also in the areas where the beam falls above 3 dB.

Regarding the RASR (Figure 20c), the calculated RASR values are shown for all of the beams defined for ECO-PRECURSOR. The thick yellow line represents the target level, while the red line indicates the minimum acceptable threshold (−17 dB). It can be observed that the RASR starts to degrade from beam 22 onward, as it becomes increasingly difficult to synthesize beams with a very narrow elevation pattern. Nevertheless, the central region of the beam still meets the RASR requirements, as specified in Table 1.

In this context, the design of the elevation beam is critical: an antenna with high or poorly controlled sidelobes may capture ambiguous signals from different ranges, increasing the RASR. Although the NESZ and RASR are distinct metrics, both are closely related to how the antenna distributes energy and rejects unwanted signals. Improving the antenna pattern—especially in elevation—not only helps reduce the RASR but also contributes to maintaining a low NESZ, thereby optimizing the SAR system’s sensitivity and image quality.

To conclude this article, a table (Table 3) comparing the four synthesized examples (strip_001, strip_010, strip_004, and strip_0.25) is included using the Chebyshev, Taylor, LSM, DE, PSO, generic GA, and GA SARAS optimization methods. In this case, we want to demonstrate whether the beams synthesized and shown in Figure 19 fulfil or do not fulfil the requirements defined in Table 1 for the proposed mission. Marked in red are the unfulfilled requirements for the mission according to Table 1.

Based on the results obtained, it can be stated that the GA SARAS genetic algorithm is the only method among those compared that consistently meets all of the specifications imposed by SAR missions, such as the minimum NESZ, AASR, RASR, and gain and the 3 dB beamwidth (BW_−3dB_). In all of the cases analyzed (strips 001, 004, 010, and 025), GA SARAS maintains NESZ values below the required threshold (e.g., −27.93 dB in Strip_001), excellent sidelobe suppression (a RASR up to −47.9 dB in Strip_010), and high gain, always above 86 dB. In addition, it achieves precise control over the beamwidth, with values in the range of 2.2° to 4.6°, which is evidence of its ability to meet complex multi-target masks. In contrast, analytical methods such as Chebyshev, Taylor, quasi-Newton, and the LSM show acceptable results for some individual parameters but fail to meet all of the requirements simultaneously.

For example, although Chebyshev achieves a good NESZ, it has worse lobe suppression (RASR), and in the LSM, despite its speed, it performs poorly on several strips. Something similar happens with the DE and PSO evolutionary methods: although they offer fast convergence and competitive results in certain cases, they fail to maintain the requirements in terms of the NESZ or RASR, with values as low as −2.9 dB for the RASR (DE, Strip_025) or with an NESZ above the acceptable limits (e.g., −10.3 dB in Strip_004). This is primarily because these algorithms, despite offering fast convergence, are not fully effective when applied to large-scale, multi-objective problems—such as the one addressed in this work, involving an array of 382 elements. This makes them unsuitable for tightly constrained applications unless reconfigured or combined with other methods. Ultimately, the ability of GA SARAS to adapt to real constraints and complex patterns makes it the most robust, efficient, and suitable solution for active array synthesis in advanced SAR systems.

## 5. Conclusions

The present work introduces several novel contributions that distinguish it from the extant research on antenna synthesis and optimization for satellite-based synthetic aperture radar (SAR) systems. Most notably, it presents the definition, implementation, and validation of SARAS (SAR Array Synthesis), a new modular MATLAB R2024a-based antenna modeling framework. SARAS employs a distinctive integration of two primary modules: the process of antenna pattern generation (APG) is employed for the purpose of analysis, whilst antenna excitation generation (AEG) is utilized for synthesis. Both modules feature intuitive graphical interfaces, enabling full customization of the antenna configurations and mission parameters. This versatility renders SARAS applicable not only to the PAZ and PRECURSOR-ECO missions but also to a wide range of satellite SAR platforms.

A key innovation is the incorporation of a custom-designed genetic algorithm (GA SARAS), tailored to the efficient optimization of complex, multi-objective, and two-way radiation masks (see Section 3.4). These masks simultaneously address stringent requirements related to the NESZ, gain, AASR, SLL, RASR, and BW_−3dB_ across a wide angular range of theta values (−90° to 90°), which are typically difficult to satisfy using the conventional methods. The cost function of the algorithm, a fundamental element of SARAS, has been developed ad hoc through the segmentation of the angular domain and subsequent optimization of the weights for each zone. This approach ensures targeted convergence towards solutions that meet both the antenna and system-level constraints (see Section 3.5).

In contradistinction to preceding studies, which relied on idealized simulations, SARAS has been validated using real measurement data from the PAZ mission. The model incorporates actual subarray patterns, antenna geometry, and TRM status data, thus bridging the gap between theoretical modeling and practical implementation.

Moreover, in order to provide a contextual framework for the method within the existing research, a range of comparisons are made with analytical approaches (for example, Chebyshev, Taylor, LSM, quasi-Newton) and evolutionary algorithms (for example, classical GA, Differential Evolution, PSO). These evaluations, conducted under identical iteration budgets, clearly demonstrate the superior adaptability of SARAS’s GA in meeting complex mission-driven multi-objective constraints (Table 3).

In the context of the large active planar arrays employed in SAR beam steering applications, genetic algorithms emerge as a particularly salient solution as a consequence of their capacity to exhibit flexibility, robustness, and the ability to manage optimization problems that are complex, constrained, or discrete in nature. While alternative methods such as Differential Evolution (DE) and Particle Swarm Optimization (PSO) may offer faster convergence, the genetic algorithm implemented by INTA (GA SARAS) provides enhanced control over beam-shaping, sidelobe suppression, and null placement, making it especially well suited to the adaptive demands of synthetic aperture radar (SAR) systems.

In summary, the present work provides a flexible, validated antenna synthesis tool and a reproducible, mission-compliant optimization strategy that significantly advances the current state of the art in SAR antenna design.

## Figures and Tables

**Figure 1 sensors-25-04835-f001:**
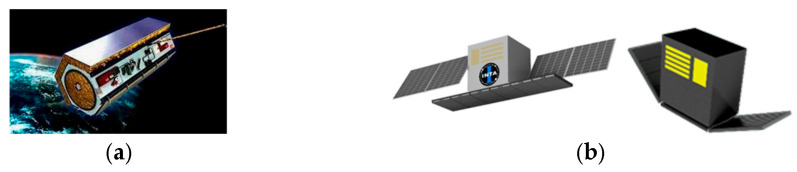
(**a**) PAZ satellite; (**b**) preliminary PRECURSOR-ECO satellite mock-up.

**Figure 2 sensors-25-04835-f002:**
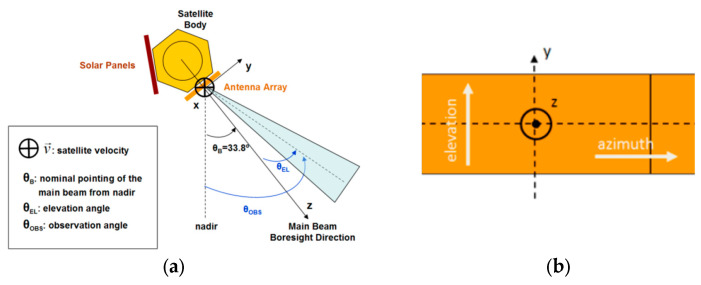
(**a**) Antenna reference system; (**b**) ECO satellite array coordinate system.

**Figure 3 sensors-25-04835-f003:**
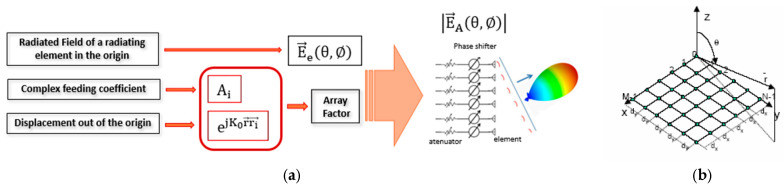
A depiction of the phased array: (**a**) equations; (**b**) axes for the rectangular distribution.

**Figure 4 sensors-25-04835-f004:**
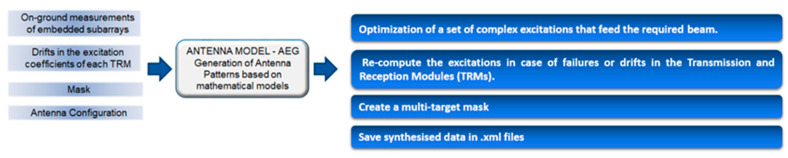
SAR array synthesis tool SARAS: functionalities.

**Figure 5 sensors-25-04835-f005:**
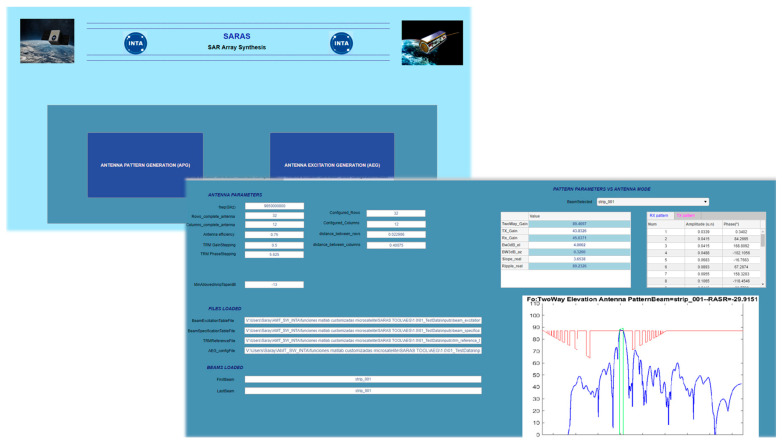
The main screen of the user interface.

**Figure 6 sensors-25-04835-f006:**
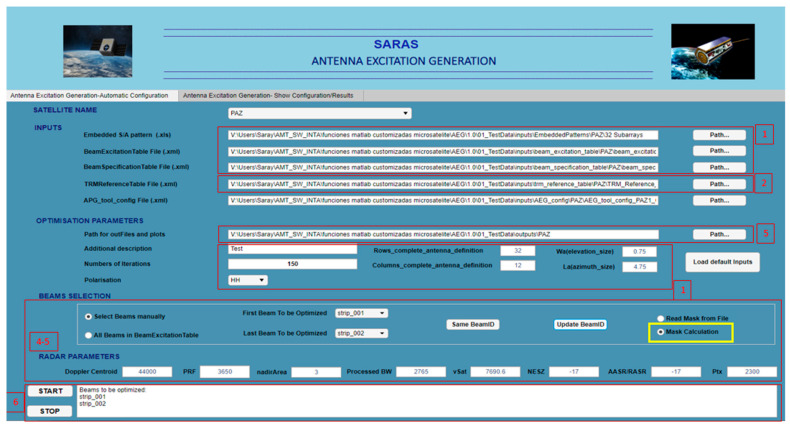
AEG module configuration screen.

**Figure 7 sensors-25-04835-f007:**
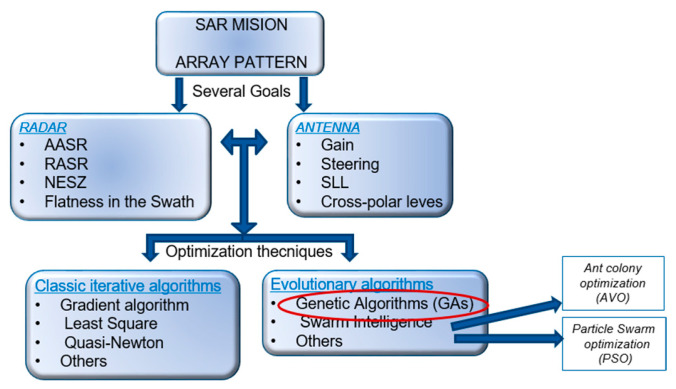
Optimization framework for phased arrays in SAR systems.

**Figure 8 sensors-25-04835-f008:**
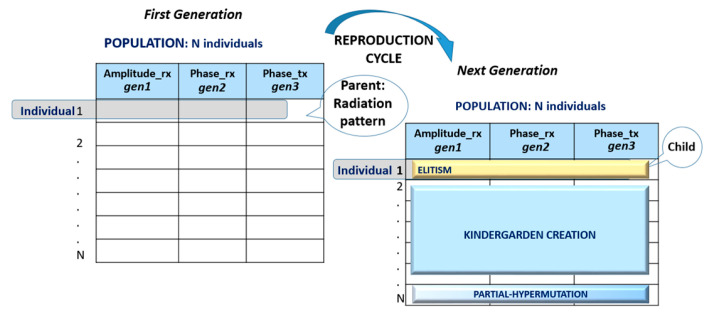
Representation of individuals in the population.

**Figure 9 sensors-25-04835-f009:**
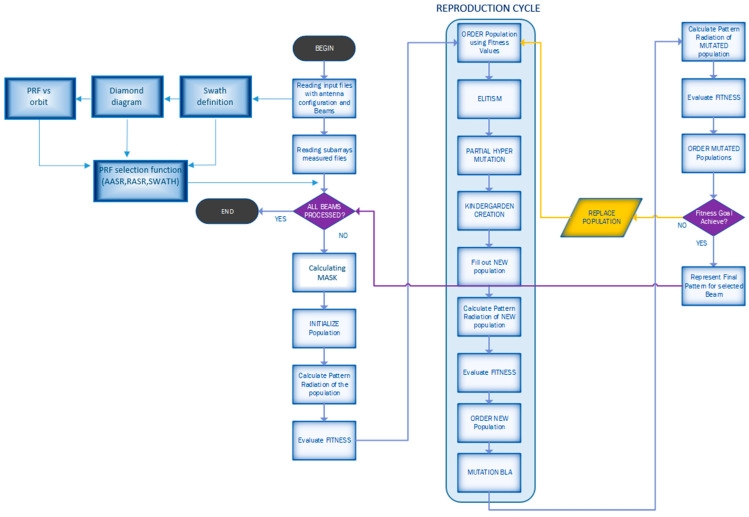
A block diagram of the genetic algorithm applied to SARAS.

**Figure 10 sensors-25-04835-f010:**
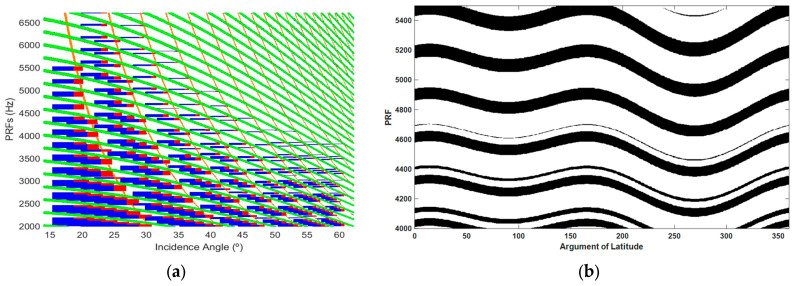
(**a**) Diamond diagram strip_001 (swath = 30 km); (**b**) PRF vs. orbit position of satellite.

**Figure 11 sensors-25-04835-f011:**
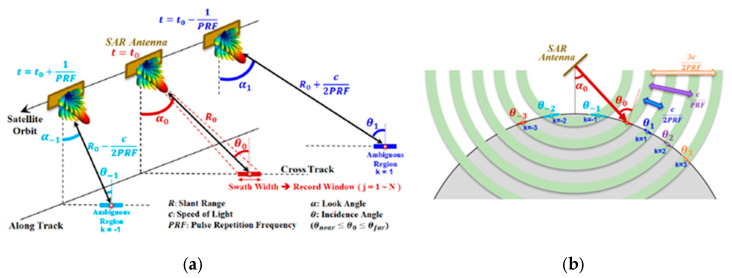
(**a**) RASR concept; (**b**) Range ambiguous regions.

**Figure 12 sensors-25-04835-f012:**
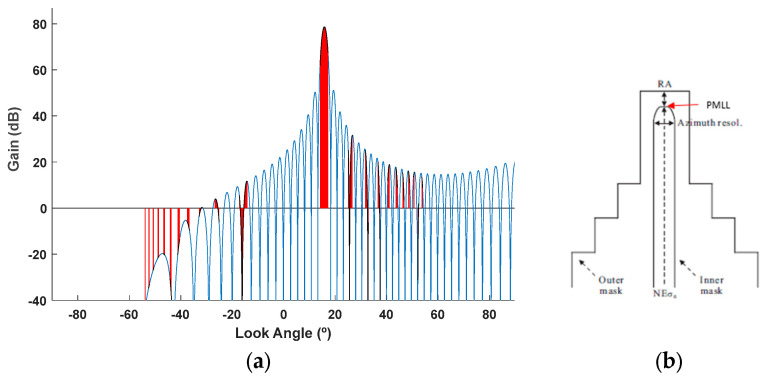
(**a**) Evaluation pattern ambiguity contribution; (**b**) Inner and outer mask.

**Figure 13 sensors-25-04835-f013:**
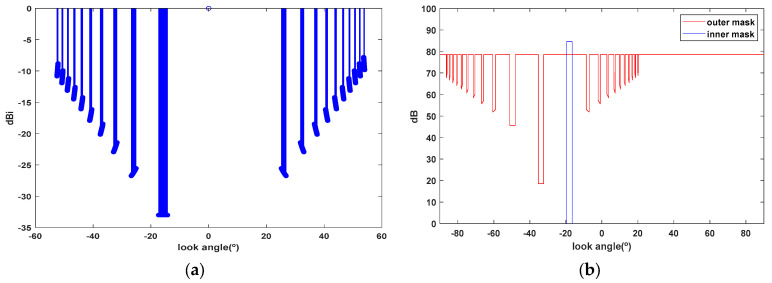
(**a**) Ambiguity regions; (**b**) final calculated mask.

**Figure 14 sensors-25-04835-f014:**
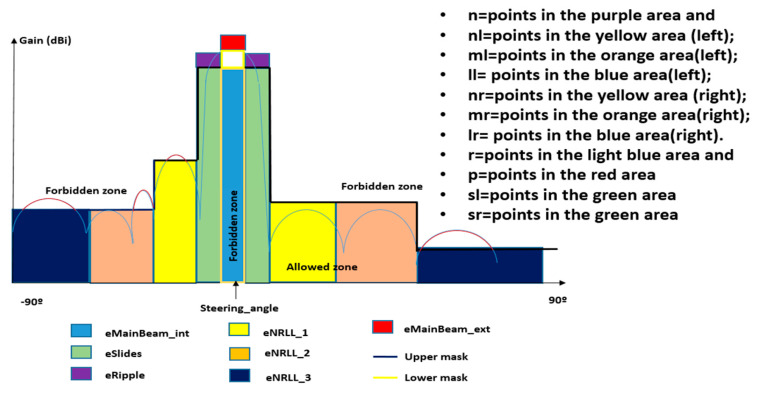
Inner mask, outer mask, and forbidden and allowed areas.

**Figure 15 sensors-25-04835-f015:**
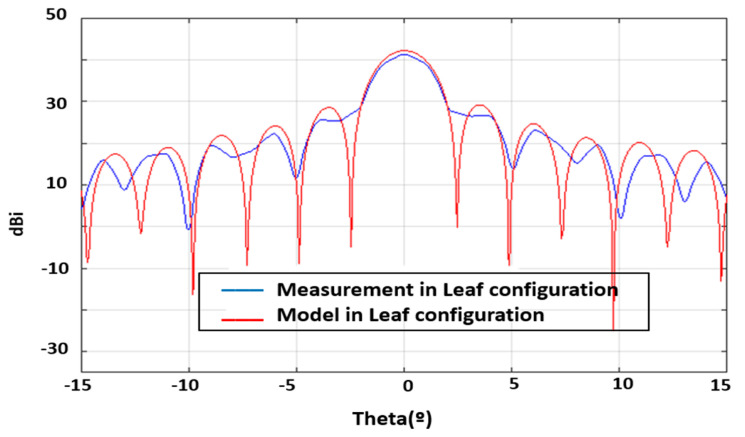
A comparison with the leaf antenna pattern measured using the PAZ satellite (SARAS-APG module).

**Figure 16 sensors-25-04835-f016:**
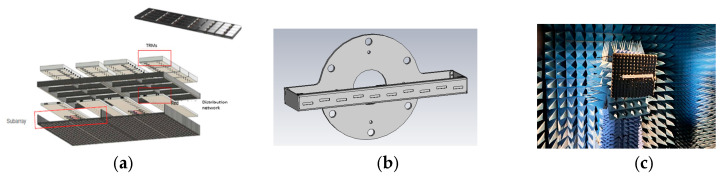
(**a**) Antenna concept; (**b**) subarray with measuring tool; (**c**) antenna measurement set-up in anechoic chamber.

**Figure 17 sensors-25-04835-f017:**
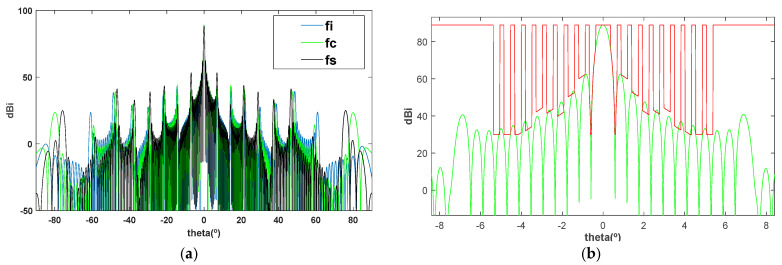
(**a**) A two-way diagram of azimuth optimization (AASR<−17 Db) in the full band; (**b**) zoom at 9.65 GHz (marked in green) according to the mask (marked in red).

**Figure 18 sensors-25-04835-f018:**
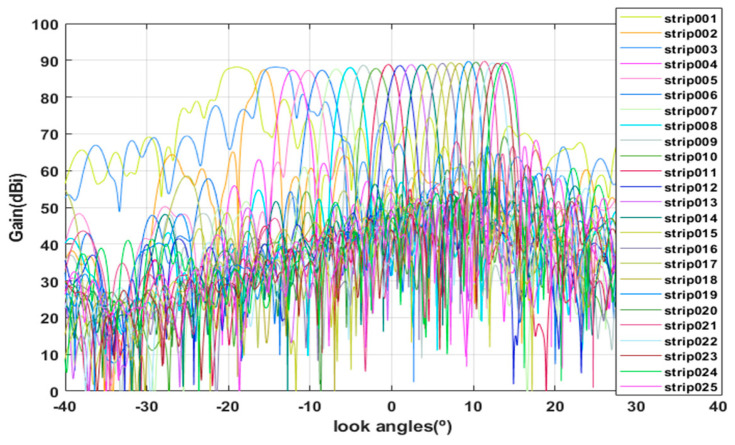
Two-way patterns synthesized with SARAS for ECO-PRECURSOR.

**Figure 19 sensors-25-04835-f019:**
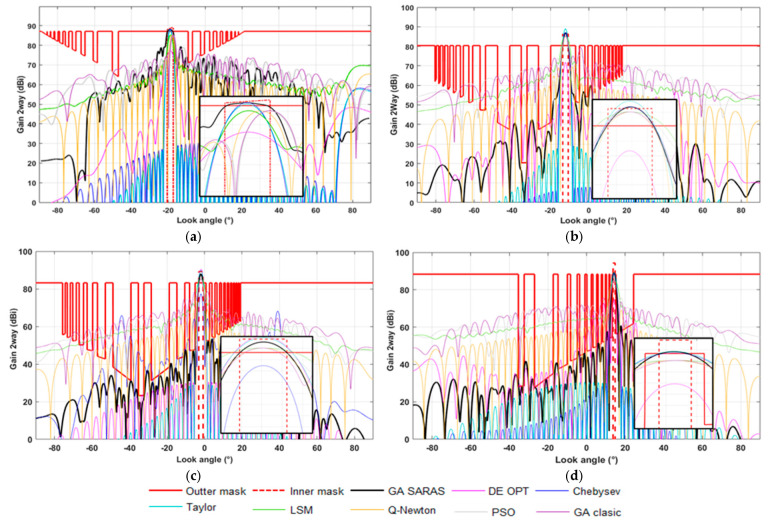
(**a**) Strip_001; (**b**) Strip_004; (**c**) Strip_010; (**d**) Strip_025.

**Figure 20 sensors-25-04835-f020:**
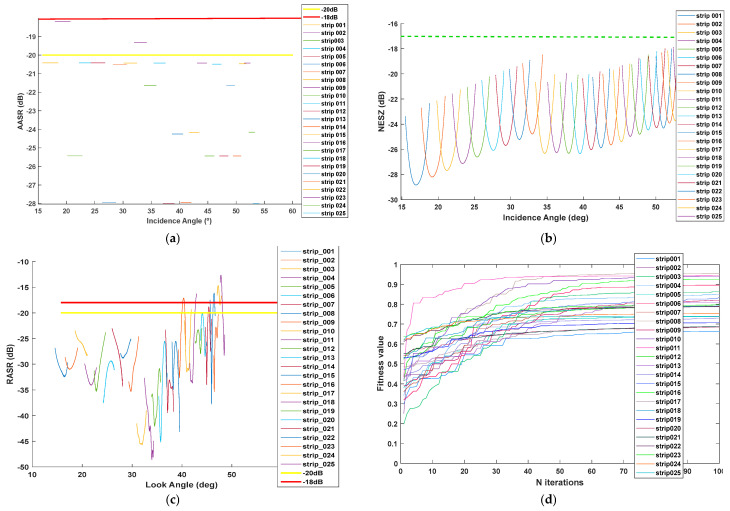
Performance analysis of PRECURSOS-ECO after pattern synthesis; (**a**) AASR for elevation beams based on target PRF; (**b**) NESZ per beam; (**c**) RASR per beam; (**d**) fitness versus iterations.

**Table 1 sensors-25-04835-t001:** Parameter values of beam Strip_01.

RASR requirement	−20 dB
Satellite altitude/roll angle	550 km/33.8°
Frequency	9.65G Hz
N elevation	32
N azimuth	12
Antena size (Wa × La)	0.75 × 3 m
Steering angle	−17.98°
NESZ	−17
Swath_BW	3000 km
PRF	3650
AASR requirement	−17 dB

**Table 2 sensors-25-04835-t002:** ECO-PRECURSOR beam definitions.

‘Beam Name’	‘Min Look Angle’	‘Max Look Angle’	‘Swath Width’	‘Target PRF’
‘strip_001’	14.26	17.35	‘30 km’	4693.31
‘strip_002’	16.56	19.07	‘25 km’	4538.24
‘strip_003’	18.30	21.24	‘30 km’	5054.86
‘strip_004’	20.25	23.10	‘30 km’	4701.88
‘strip_005’	22.14	24.90	‘30 km’	4693.31
‘strip_010’	30.72	33.03	‘30 km’	5017.93
‘strip_025’	47.64	48.57	‘22 km’	5296.56

**Table 3 sensors-25-04835-t003:** A comparison of the mission parameters obtained through the synthesis of different methods.

	‘PARAM’	GA SARAS	CHEBY	TAYLOR	QNEWTON	LSM	DE	PSO
Strip_001	NESZ_min (dB)	−27.93	−29	−28.4	−25.6	−25.9	−18	−25.4
	AASR (dB)	−19.39	−19.39	−19.39	−19.39	−19.39	−19.39	−19.39
	RASR (dB)	−44.5	−45.7	−32.63	−31	−22.1	−35.3	−14.4
	GAIN (dBi)	88.2	88.2	87.6	85.2	85.7	77.2	85.4
	BW −3dB (°)	4.6	2.7	2.8	2.7	3.3	4.95	2.87
Strip_004	NESZ_min(dB)	−25.5	−25.6	−27.6	−23.8	−23.9	−10.3	−23.8
	AASR (dB)	−19.42	−19.42	−19.42	−19.42	−19.42	−19.42	−19.42
	RASR (dB)	−40.3	−44.2	−52.25	−29.2	−9.8	−12.7	−17.4
	GAIN (dBi)	86.9	87	88.9	85.1	85.21	71.7	85
	BW −3dB(°)	2.85	2.9	2.72	3	4.7	2.23	3.77
Strip_010	NESZ_min(dB)	−25.4	−14.9	−24.4	−22.8	−22.8	−27.9	−22.7
	AASR (dB)	−21.3	−21.3	−21.3	−21.3	−21.3	−21.3	−21.3
	RASR (dB)	−47.9	−27.2	−45	−26.5	−13.6	−36.5	−12.8
	GAIN (dBi)	87.9	77.3	89.8	85.1	85.2	90.4	85.1
	BW −3dB(°)	2.5	2	2.6	3	4.7	2.6	3.4
Strip_025	NESZ_min	−21.6	−21.1	−21.6	−17.7	−17.7	−7.11	−17.7
	AASR (dB)	−27.42	−27.42	−27.42	−27.42	−27.42	−27.42	−27.42
	RASR (dB)	−34.9	−42	−45.25	−15.1	−7.1	−2.9	−5.8
	GAIN (dBi)	89.1	88.6	89.2	85.3	85.21	77.7	85.2
	BW −3dB(°)	2.2	2.74	2.7	2.14	4.7	1.66	3.2

## Data Availability

No new data were created or analyzed in this study.

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
