# Peer review of "Antenna Model with Pattern Optimization Based on Genetic Algorithm for Satellite-Based SAR Mission"

_sensors, 2025, doi:10.3390/s25154835_

Round 1

Reviewer 1 Report

Comments and Suggestions for Authors

This manuscript presents antenna model with pattern optimization based on genetic algorithm for satellite-based SAR mission. There exists 4 major issues and 1 minor issue. Comments are listed as follows.

Major issue 1: Authors should summarize main contributions of this manuscript in the part of introduction.

Major issue 2: This manuscript is like an experiment report rather than a scholar papers. Authors should read relevant scholar papers and improve this manuscript.

Major issue 3:What’s the novelty of this manuscript?

Major issue 4:Authors should compare the proposed method with other pattern optimization methods. At least one comparison method should be proposed within 5 years.

Minor issue 1: Font size in Figures 5,6, 7, 10, 11, 12, 14, 15,16,17,18,19,20,21,23,24,26,27 is too small for readers. Authors should enlarge font size in these figures.

Reviewer 2 Report

Comments and Suggestions for Authors

Thanks to the authors for introducing their tool to optimize antenna patterns for satellite SAR.

This paper is well written.  But it includes too many background introductions and much of the content focuses on the report on the built software.  To a scientific paper, readers do not care too much about the software you did. They only want to know the antenna model and how to do the optimization. From this point of view, I suggest authors to delete the descriptions of the software, and use more formulas to describe your model and the optimization process. The flowchart is appreciated.  From my view, 20 pages are enough to illustrate the entire story. 

Please concentrate on the method you used, the examples you used to test your method, and the results comparison with others, and so on. Some suggestions:

The quality of the figures should be improved, too.

For clarity and depth, please elaborate on the specific functions of each module. Discuss the algorithms used for analyzing patterns in APG and the algorithms used for synthesizing patterns in AEG. This should include any unique features or advantages provided by the module compared to existing methods.

Carefully discuss the strategies implemented to optimize, for example, the repetition rate setting, balancing azimuth and distance ambiguity. Provide technical justification for the preferred method or threshold to ensure optimal mode synthesis across different scenarios.

Round 2

Reviewer 1 Report

Comments and Suggestions for Authors

After revision, this manuscript has been improved. There still exists several Minor issues in this manuscript.

Minor issue 1:  Authors should add references which are corresponding to comparison methods in this manuscript.

Minor issue 2:Authors should list the summarized main contribution in the part of introduction.

Reviewer 2 Report

Comments and Suggestions for Authors

No more comments.

Please make the figures more readable. For example, the quality of Fig. 15.
